# Topic Break Detection in Interview Dialogues Using Sentence Embedding of Utterance and Speech Intention Based on Multitask Neural Networks

**DOI:** 10.3390/s22020694

**Published:** 2022-01-17

**Authors:** Kazuyuki Matsumoto, Manabu Sasayama, Taiga Kirihara

**Affiliations:** 1Graduate School of Sciences and Technology for Innovation, Tokushima University, 2-1 Minamijosanjima-cho, Tokushima-shi 770-8506, Japan; c612035024@tokushima-u.ac.jp; 2Department of Information Engineering, National Institute of Technology, Kagawa College, 551 Kohda, Takuma-cho, Mitoyo-shi 769-1192, Japan

**Keywords:** interview dialogue, topic segmentation, dialogue analysis

## Abstract

Currently, task-oriented dialogue systems that perform specific tasks based on dialogue are widely used. Moreover, research and development of non-task-oriented dialogue systems are also actively conducted. One of the problems with these systems is that it is difficult to switch topics naturally. In this study, we focus on interview dialogue systems. In an interview dialogue, the dialogue system can take the initiative as an interviewer. The main task of an interview dialogue system is to obtain information about the interviewee via dialogue and to assist this individual in understanding his or her personality and strengths. In order to accomplish this task, the system needs to be flexible and appropriate for detecting topic switching and topic breaks. Given that topic switching tends to be more ambiguous in interview dialogues than in task-oriented dialogues, existing topic modeling methods that determine topic breaks based only on relationships and similarities between words are likely to fail. In this study, we propose a method for detecting topic breaks in dialogue to achieve flexible topic switching in interview dialogue systems. The proposed method is based on multi-task learning neural network that uses embedded representations of sentences to understand the context of the text and utilizes the intention of an utterance as a feature. In multi-task learning, not only topic breaks but also the intention associated with the utterance and the speaker are targets of prediction. The results of our evaluation experiments show that using utterance intentions as features improves the accuracy of topic separation estimation compared to the baseline model.

## 1. Introduction

Robots that can interact with humans are becoming commonplace in everyday life. For example, robots that are capable of task-oriented dialogues, such as responding to inquiries in different circumstances such as corporate and hotel receptions and restaurant reservations [1,2,3,4,5,6], are practical and in high demand. However, in the case of non-task-oriented dialogue systems, the accomplishment of a specific task is not the primary objective. Instead, the goal is free and flexible dialogue. Such systems are called chatting dialogue systems [7], and with the recent development of artificial intelligence technology, research on generating response sentences is actively being conducted.

However, since the topics of general conversations are more diverse than those of task-based dialogues, it is very difficult to understand the intention of an utterance and to generate a response accordingly. For this reason, in conventional research on continuing chatting, when the content and the intention associated with the utterance cannot be understood, many dialogue models return safe responses that can be applied in any situation.

In addition, an objective evaluation of the generated responses is difficult to achieve in general. In particular, in daily conversations between people who have close relationships, there are dialogues in which subjects, objects, abbreviations, etc., are frequently used and can be established only when there are close relationships, such as between family and friends. For example, consider the sentence “I had a delicious sandwich in front of the station the other day.” If the dialogue robot is unaware of the user’s daily activities, it may not be able to provide an appropriate response. It is difficult for task-based systems and existing dialogue models that focus only on continuing a conversation to appropriately deal with these utterances. Since these dialogues between closely related individuals cannot be understood by a third party on the basis that there is a relation between the utterance and the response, it is difficult to grasp the intention associated with the utterance.

In this study, we focus on interview-style dialogues. We assume that the dialogue system plays the role of the interviewer, and another user (human) serves as the interviewee. In an interview dialogue, the interviewer can control the dialogue, and even though the topic is broad, the dialogue can be facilitated by limiting the questions to those tailored to the other person. The interviewer first understands the user’s speech intention. In addition, the interviewer generates responses to additional questions and topics based on the understood speech intentions. Based on this process, the dialogue system can take the initiative while allowing the user to speak freely. Moreover, by not limiting the scope of the topic, the system can obtain information other than what it wants to ask the user in advance, which has the effect of continuing the dialogue.

Furthermore, regarding the problem of a disinterested user, if the information acquired in the previous interview is kept in the dialogue’s history and can be referred to in a later conversation, the same topics can be avoided as much as possible, and the system can take the initiative to ask new questions and discover new interests. In this manner, the interview dialogue format can elicit the other party’s actions, thoughts, and experiences more efficiently. Moreover, the intention associated with the utterance, which is important for smooth dialogue, can be grasped more accurately.

The purpose of this study is to identify topic breaks in dialogues during an interview. If the dialogue can be segmented into topics, it will be easier for the dialogue system, which is the interviewer, to delve deeper into the topic. We also considered that by referring to past topic units, it would be easier to control the flow of the dialogue by avoiding similar topics as much as possible, successfully diverting past topics and so on. In this study, we used an interview dialogue corpus in which the intentions of the interviewer or interviewee were manually tagged, and each utterance was assigned an originally created utterance intention tag. By targeting this corpus and determining the occurrence of the utterance intention tags and their relationship to the flow of the utterance, it was possible to detect breaks in the topic. We also believe that it can be used as a reference for interviewers to intentionally change topics. 

Most existing studies have attempted to detect topic boundaries using topic modeling or word association. However, in a dialogue wherein the topics are related to each other and transition smoothly, such as in an interview dialogue, it is difficult to identify the transition point of the topic by using methods such as topic modeling based on word occurrence probability.

The proposed method uses distributed representations of sentences (sentence embedding) that can account for context, such as BERT, as features for detecting topic breaks. In order to extend features, we propose a model that considers speech intention and a model that uses sensory features. This method enables the dialogue system to determine the appropriate topic break when a topic is presented to the interviewee in an interview dialogue and for the identification of topic breaks by the interviewee. This enables the system to conduct interview dialogues seamlessly and continuously and to alert the interviewees of their potential through dialogue.

In Section 2, we present examples of related studies on dialogue systems and topic segmentation in dialogues and discuss the differences between them and this study. Section 3 provides an overview and analysis of the interview dialogue corpus used in this study. In Section 4, we explain the proposed method and the topic segmentation model. Section 5 describes the evaluation experiment of the proposed method, and Section 6 discusses the results of the experiment. In Section 7, we summarize the study and discuss future work.

## 2. Related Works

This section presents examples of related research on dialogue systems and previous research on topic segmentation and describes the differences when compared to our research.

### 2.1. Studies on Dialogue System

The MAD proposed by Zhang et al. [8] is a new memory expansion dialogue management model that adopts the structure of a memory controller, slot value memory, and external memory. In MAD, the dialogue state is tracked by managing the value of the semantic slot (cooking, price, location, etc.) using the slot value memory. In addition, external memory expands the representation of the hidden layer based on the conventional recurrent neural network by storing context information. This model exceeded the baseline in the evaluation experiment.

In addition, there is ongoing research on dialogue systems, and many researchers [9,10,11,12,13,14,15,16,17,18,19,20,21,22,23,24,25,26,27,28,29,30,31,32,33] and IT companies (https://github.com/microsoft/botframework-sdk, accessed on 16 January 2022, https://developers.facebook.com/docs/messenger-platform/, accessed on 16 January 2022, https://docs.sebastien.ai/ accessed on 16 January 2022) are currently focusing on research and development involving chat dialogue AI.

In order to realize a non-task-oriented chatting dialogue, it is necessary to consider not only speech intention and semantic understanding, but also contextual understanding and world knowledge. Many studies have attempted to generate utterances or to understand the topic of an utterance. In our study, we propose a method for detecting topic breaks in system-driven interview dialogues.

### 2.2. Studies on Topic Segmentation

In previous studies, Miyamura et al. [34] proposed a method for detecting topic transitions by using a lexical chain. The approach first divides the corpus into arbitrary window widths using a technique called DiaSeg. Subsequently, topic transition is detected by examining the connection of each window using a lexical chain. The lexical chain refers to the connection of vocabulary using nouns, verbs, and adjectives and classifies topics based on the connections between these parts of speech. In their study, topic breaks were detected according to the difference in similarity by vectorizing utterance sentences instead of lexical chaining. In a dialogue wherein the interviewer always has the initiative, such as an interview dialogue, it is not necessary to detect the dialogue initiative as in DiaSeg proposed by Miyamura et al. for a chat dialogue. However, in their study, it was determined that it is effective to extract candidate topic boundaries based on the presence or absence of dialogue initiative. Therefore, we believe that the determination of whether the speaker is an interviewer is sufficiently effective as a feature of topic segmentation in our study.

In addition, Tomiyama et al. [35] proposed a method for detecting topic transitions using multimodal features. They proposed a technique for detecting topic transitions by considering speech spectra and speaker movements in addition to language features. The author aimed to detect topic breaks by extracting the voice spectrum and detecting the differences between speakers. In our research, we use the transcribed text (subtitles) of a dialogue program as training data for developing an interview dialogue system. In order to utilize multimodal information, it is necessary to use audio and images of the interviewer and interviewee, as well as the text. In order to obtain useful multimodal data, it is more effective to use unedited data as much as possible. The dialogue program of interest is similar to a live dialogue in that it is conducted with detailed discussions beforehand; however, it is not edited. It differs from the recording of a normal dialogue scene because the speaker’s voice is sometimes played without the speaker being onscreen. For these reasons, we believe that it is more effective to use only linguistic data that are less affected by data loss, as described above.

Tanaka [36] detected topic breaks based on interjections and conjunctions. In this study, topic breaks are detected using words such as “but” and “maybe” to break topics. The entire sentence is vectorized, but if these features can be extracted, the process of topic segmentation can be conducted more effectively. The inspirational words and conjunctions of interest in their study are characteristic of the interviewers’ speech. For example, “By the way,” “Sate,” and other words are often used to switch topics. These are rarely used by the interviewee and are more prevalent in the interviewer’s speech.

Arguello et al. [37] proposed a topic segmentation method that combines the features of topic shifts based on lexical cohesion and linguistic features, such as the syntactically distinct features of the first and last utterances of a segment. In their report, they proposed a supervised learning algorithm for dialogue topic segmentation called Muesli, which combined linguistic features that indicate contribution functionality with local context indicators. The experimental results demonstrate that using dialogue as the basic unit of discourse, rather than contributions, significantly improves performance in topic segmentation. Their research is similar to that of Miyamura et al.’s DiaSeg in that they identify topic transitions using lexical association. However, they focused on the fact that the vocabulary used in dialogue differs depending on the role of the speaker (student, teacher, etc.), and they trained models for each role. In our research, since the roles of the interviewer and interviewee are distinct, we prepared speech intention tags for each role in the corpus.

Topic segmentation for dialogue text is more difficult than for general documents. Existing neural models for topic segmentation are usually built on recurrent neural networks (RNNs) and convolutional neural networks, which work well when applied to written text but perform poorly for interactive text (e.g., when colloquial language is used). Therefore, Zhang et al. [38] formulated topic segmentation as a sequence labeling task to improve the semantic expression of sentences. If we use sequence labeling tasks together with RNNs and CNNs as they did, large-scale labeled data are required. The problem is that the corpus that we use is not large, and dialogue topics and the interviewees’ attributes are diverse. It is possible to extend dialogue data using similar vocabulary, but in general, linguistic information is more difficult to extend automatically compared to images or speech. If the meaning of a word changes even slightly, the overall meaning may change significantly, and there is a distinct possibility that simple data expansion will result in noisy data. In this study, we propose a method for constructing a topic segmentation model by resampling a small amount of data without using simple data expansion with semantic changes.

Sheikh et al. [39] proposed a new approach for topic segmentation in speech recognition transcripts by measuring vocabulary coupling using bidirectional recurrent neural networks (Bidirectional RNNs). This approach does not use segmented corpora or pseudo-topic labels for learning. The model learns using news articles obtained from the internet. An evaluation using the ASR (automatic speech recognition) transcript of a French television news program demonstrated the effectiveness of the proposed method. Text delimitation cannot be used when topic segmentation is implemented for the transcription of audio data. In this study, the target problem is different because the data have already been transcribed into text.

Kimura et al. [40] proposed a system that allows users to change topics appropriately in a manner that is not awkward, even if the topic is interrupted. They built a system that presents topic candidates related to the topic being discussed so that even people who are not adept at dialogue can suggest topics seamlessly. They used TF-IDF to analyze time-segmented documents and detect important words. MeCab was used for word segmentation. In addition, as an appropriate method to present topics, we searched for headlines of news articles and presented them as topic candidates. Consequently, it was concluded that approximately 70% of the topics were consistent when the top two were allowed. In general, the problem with TFIDF is that it cannot confront unknown vocabularies, because words that do not appear in the training data are treated as unknown and cannot be used as meaningful information in TFIDF calculations. In the case of language models trained on large language data, such as BERT, the number of unknown words is small, and even if unknown expressions are present, they are robust against unknown words because they can infer distributed expressions based on associated lexical information. In this study, we use pre-trained BERT to obtain features to address the size of the target corpus and a wide variety of topics.

Faycal et al. [41] proposed a deep neural model with a multi-hop attention mechanism (MHOPSA-SEG) to acquire multifaceted semantic information for topic segmentation of text. In this model, at each step of the attention mechanism, different weights are assigned to words according to the weight of the previous memory step. This method facilitates the handling of different semantic vector representations in multiple sentences.

In their method, features from pre-trained BERT are used, and complex networks such as Bi-LSTM, highway network layer, attention layer, and multihop attention layer are trained to perform binary classification of topic separation. The prediction target is binary, which is the topic break, but using a network that is too complex may lead to overtraining of the model. In addition, the target is an article such as Wikipedia, which is considered to be more prone to topic separation than dialogue data. In our research, we aimed to find the most suitable features and algorithms for topic segmentation by using multiple conditions without making the network structure too complex.

Solbiati et al. [42] proposed a topic segmentation method for transcribing meetings with multiple speakers using the embedded representation of BERT. Topic detection is a difficult task in meeting transcription because multiple participants speak freely, and topics are intermingled. In general, automatic topic segmentation using language modeling requires a large dataset. Given that preparing a large corpus with teacher labels is very costly, unsupervised methods such as latent topic modeling methods are required. However, methods such as LDA target written text with few ellipses and ambiguous expressions, making it difficult to deal with spoken text such as transcriptions of meetings. They used a pre-trained BERT model to improve the accuracy of conventional unsupervised topic segmentation methods. As a result, their proposed method achieved an error rate reduction of 15.5% compared to existing methods. Dialogue data used in our study constitute a dialogue between an interviewer and an interviewee. However, unlike a conference, it is a chatty dialogue without a clear subject, and the related topics often transition smoothly. Therefore, the interviewer’s speech, which takes the initiative in the dialogue, is likely to trigger a topic change. In this study, considering this characteristic, we focus not only on the utterance itself but also on the speaker’s information (interviewer or not).

Other studies include the determination of dialogue boundaries [43], a method for estimating utterance topics for conference support [44], and bilingual topic modeling [45]. However, their approach differs from ours, which is based on an originally constructed interview dialogue corpus. Topic modeling and word similarity, which are often employed in traditional methods, are not suitable for interview dialogues wherein topic transitions are ambiguous and gradual. For example, in interview-style dialogues, such as chatting, there are few transitions from the topic of cars to a completely different topic. Instead, many words related to each other tend to appear in adjacent topic segments, such as the transition to the topic of obtaining a car license. In such a case, it becomes difficult to detect the correct topic based on topic modeling and commonly occurring vocabulary.

## 3. Interview Dialogue Corpus

This section describes the interview dialogue corpus used to apply the proposed method. Specifically, the details of the data collection, annotation of the speech intention tags, and corpus analysis are explained.

### 3.1. Data Collection

Sentences uttered by interviewers and guests (interviewees) were extracted from the subtitled text of dialogues in the TV program “Tetsuko’s Room.” A speech intention tag was assigned to each utterance by determining the intention associated with the utterance. The first advantage of using data from dialogue programs is the high reliability of the content, since the subtitles are created by a specialized company. Furthermore, in this program, the direction of the topic is decided in detail in advance, and the interviewee is familiar with the dialogue due to the hosting of the same program for many years. Therefore, it is easy for a third party to understand. Finally, the size of the dataset is large because “Tetsuko’s room” is a long-running program.

### 3.2. Speech Intention Tag Annotation

In a study by Sasayama et al. [46,47,48,49], an interview dialogue corpus of dialogue programs (30 min each) from 8 September 2017, to 29 September 2019, was constructed. There is only one interviewer, and the number of interviewees (“guests”) ranges from one to three. The types of speech intention tags are listed in Table 1.

The total number of the eight main tags assigned by the annotator in the interview dialogue corpus by the interviewer and interviewee is shown in Table 2. The most frequent tag among the tags assigned by multiple annotators was chosen as the correct number. In the case of a tie, the tags were selected in alphabetical order. The corpus was annotated with speech intention tags by three annotators, and Kappa statistics were computed, yielding values of 0.72 for the main tags (eight types) and 0.55 for the sub-tags (18 kinds), indicating a good agreement. The agreement rate of speech intention tags was also calculated, 94.2% for the main tag and 64.6% for the sub-tags, when the number of tags that matched among the three annotators was considered to be in agreement.

The topic breaks to be predicted in this study were manually assigned to all 30 dialogues. One worker assigned the topic breaks, and two researchers checked the annotations and made decisions. Specifically, only those tagged by the annotators and approved by the two evaluators were adopted as official topic breaks. The breakdown of topic breaks for each interviewer and interviewee is as shown in Table 3. 

### 3.3. Corpus Analysis Based on TF-IDF Document Vector

Here, we investigate how the similarity between adjacent topics can be obtained by documenting the set of intra-topic utterances based on tf-idf for a corpus of interview dialogues with topic breaks.

Equation (1) shows the formula for calculating importance $fidf_{t}^{d}$ of a word t in a topic interval *d*. Equation (2) shows the formula for calculating reverse document frequency $idf_{t}^{d}$ of a word t in a topic interval *d*. $tf_{t}^{d}$ is the frequency of occurrence of word *t* in a topic interval *d*. *N* is the total number of topic intervals. The base of the logarithms was 10.
(1)$$tfidf_{t}^{d}=tf_{t}^{d}\times idf_{t}^{d}$$
(2)$$idf_{t}^{d}=\log\frac{N}{df_{t}^{}}+1$$

We create a bag-of-words vector (tf-idf vector) for each topic section, wherein the dimension is the word ID, and the value is the importance ascribed by tf-idf. Given that the total number of words in the interview dialogue corpus is 6662, the number of dimensions becomes a vector of 6662 dimensions.

The cosine similarity between the tf-idf vectors *T_x_* and *T_y_* in a topic is given by Equation (3). $\left|T_{x}\right|$ and $\left|T_{x}\right|$ represent the vector size of each tf-idf vector.
(3)$$sim(T_{x},T_{y})=\frac{T_{x}\cdot T_{y}}{\left|T_{x}\right|\times \left|T_{y}\right|}$$

The distributions of similarity between tf-idf vectors and all tf-idf vectors in adjacent topic segments are shown in Figure 1. Table 4 lists the common word sets for the top 10 similarities.

When similarity is relatively high, there is a large percentage of multiple occurrences of the same word. Figure 1 shows the distribution of similarity among adjacent topic segments and among all segments, and there is a degree of similarity above 0.2. In addition, since the number of words contained in a sentence in a dialogue is not large, it is considered that there is no significant variation in the similarity if only one sentence is shifted back and forth. This indicates that it is quite difficult to determine the delimitation point by simply setting the similarity threshold. In other words, using only document similarity by document vector based on word importance by tf-idf is inadequate for the effective detection of a topic break.

## 4. Topic Segmentation Model

In this research study, we performed topic segmentation based on the features of each uttered sentence. In the past, topic analysis has often been based on probabilistic methods such as latent Dirichlet allocation (LDA), which classifies or clusters documents into topics based on word co-occurrence or the probability of occurrence based on a large amount of text data. In the case of LDA, documents are classified into topics or clustered based on word co-occurrence or the probability of occurrence based on large amounts of text data. However, in interview dialogues, which are the subject of this study, the amount of data in a single utterance is usually not sufficient for obtaining the probability of word occurrence from the utterances of the interviewer and the interviewee. As such, it is expected that topic analysis using LDA will be difficult. Therefore, in this study, we considered that by transforming sentences into vector representations that capture context and meaning, we could efficiently and accurately grasp the relationships between utterances in a dialogue.

Traditionally, methods such as bag-of-words and contextual co-occurrence vectors have been used to convert sentences into semantic vectors. However, in recent years, distributed representations of words and sentences have often been used. In the case of distributed representation, words and sentences can be represented as dense real-valued vectors by embedding them in a high-dimensional real-valued vector space, and relationships (such as similarity) can be expressed numerically. The most famous distributed representations of words are Word2vec [50], fastText [51], and GloVe [52]. For the distributed representation of sentences, methods such as Doc2Vec (Paragraph2Vec) [53], which is an extension of word2vec to documents, and bidirectional encoder representations from transformers (BERT) [54], which use transformer networks, are often used. BERT can obtain distributed representations of words in each context. BERT can be used to obtain distributed representations of words by context and to obtain distributed representations by sentences or paragraphs, which can be used for a variety of tasks.

In the following subsections, we describe the vectorization of utterances in dialogues using BERT and its improved model and the topic segmentation method based on this approach.

### 4.1. Vectorization of Dialogue Utterance Text

In this study, we used a pre-trained model of BERT, its improved version DistilBERT [55], and the pre-trained model of Sentence BERT [56], which is specialized for sentence clustering to vectorize the utterance text in a dialogue. Since these models have been pre-trained on large-scale Japanese sentence data such as Wikipedia articles, they are expected to produce context-aware feature vectors.

The results of some BERTs trained on Japanese corpora have been published by the Kurohashi Laboratory at Kyoto University (https://nlp.ist.i.kyoto-u.ac.jp/?ku_bert_japanese, accessed on 16 January 2022), Inui Laboratory, at Tohoku University (https://github.com/cl-tohoku/bert-japanese, accessed on 16 January 2022), NICT (https://alaginrc.nict.go.jp/nict-bert/index.html, accessed on 16 January 2022), and others. In this study, we used the pre-trained model published by NICT and 768-dimensional CLS tokens just before the final layer as feature vectors. This pre-trained BERT model is distributed using SentencePiece tokenization and byte pair encoding (BPE) to reduce the vocabulary size. In this study, the model based on BPE was used for the feature extraction of utterances. For Sentence BERT, we used a pre-trained model published by Nittetsu Solution (NSSOL) (https://huggingface.co/sonoisa/sentence-bert-base-ja-mean-tokens, accessed on 16 January 2022). For Sentence BERT, we use the pre-trained model published by NSSOL (Nittetsu Solution), and for Distill BERT, we use Laboro DistilBERT Japanese (https://github.com/laboroai/Laboro-DistilBERT-Japanese, accessed on 16 January 2022).

The feature vectors used in this study, BERT, DistilBERT, and Sentence BERT, were compressed into two dimensions and visualized to investigate the possibility of classification. Three-dimensional compression algorithms, UMAP [57], t-SNE [58], and PCA, were used for visualization. The results are shown in Figure 2. BERT indicates BERT-CLS, SBERT indicates Sentence BERT, and DBERT indicates DistilBERT.

In Figure 2, we classify data by the type of combination of speaker and topic labels, and random undersampling is used. The red dots represent topic breaks, and the green dots represent non-topic breaks. From this figure, it can be observed that topic breaks and non-topic breaks do not form distinct clusters, and it may be difficult to separate them into two classes based on feature vectors only.

### 4.2. Topic Segmentation Model Based on Neural Networks

In our research, we applied a forward propagation neural network with one or more inputs as the architecture for learning topic breaks. The inputs are the sentence vectors of a single utterance or two consecutive utterances. Multiple sentence vectors were used as inputs because of the following. In the interview dialogue, the interviewer asked questions to the interviewee, and the interviewee answered them. As such, contextual information, such as how the interviewee responded to the question and how the interviewer responded to the answer, is considered to be effective in detecting topic breaks. In other words, if we use the immediately preceding utterance as the context of the dialogue, it is possible to discriminate dialogue to some extent.

In the case of the topic segmentation model with two utterances as the input, to identify the relationship between before and after the topic break, a three-valued classification model was used to classify which of two consecutive utterances was the topic break and which was not.

Figure 3a,b show examples of input features for binary and three-valued classifications.

In the case of a model in which only one sentence is used as the input and the classification is based on whether it is a topic break or not, a sentence that is almost the same may not be judged to be a topic break depending on its relation to preceding or succeeding sentences. We do not accept more than three sentences as inputs because if there are too many prediction patterns (four patterns: nnn, tnn, ntn, and nnt in the case of three sentences), there will be insufficient training data to determine the segmentation point for each pattern, and overtraining will occur.

It is also possible to use recurrent neural networks to learn each context using N vectors of utterances before the target utterance. However, in such a case, if the context contains multiple topic breaks, the prediction may become more difficult. For this reason, we use only one input context and train the topic break labels of the context together to increase the amount of information that can be used as cues.

A dialogue system that plays the role of an interviewer can at least determine in advance the speech intentions to which its utterances correspond. Some of the intentions are important cues in determining a topic break, such as a question to change the topic. For this reason, the proposed model improves the accuracy of topic break estimation by providing the interviewer’s speech intention tag as an input only when the input utterance contains the interviewer’s utterance.

As previously indicated, there are three levels of speech intention tags: 8, 11, and 3. Each of these vectors is used as a one-hot vector, and the three vectors are concatenated to form a 22-dimensional speech intention vector. This vector is not only used as the input, but also as a prediction target in the interviewee’s speech, which is expected to improve learning efficiency. In the model with two sentences, only the interviewer’s speech intention vector is used as the input. However, the interviewee is a 22-dimensional vector of zeros that is used to specify one of the two sentences. In other words, the input is a 44-dimensional vector consisting of speech intention vectors of the two sentences concatenated back-to-back.

In the case of binary classification, a vector series of embedding of the input dialogue sentences $S=\left\{S_{1},\;S_{2},\ldots ,S_{M-1},S_{M}\right\}$ can be represented as $S^{\prime }=\left\{\left\{S_{1},T_{1,}\right\},\;\left\{S_{2},T_{2}\right\},\ldots \left\{S_{M-1},T_{M-1}\right\},\;\left\{S_{M},T_{M}\right\}\right\}$ if the input is supplemented with a speech intention tag T. Furthermore, if we add a vector U representing the speaker type (interviewee or interviewer), it can be represented as $S^{\prime \prime }=\left\{\left\{S_{1},T_{1,},U_{1}\right\},\;\left\{S_{2},T_{2},U_{2}\right\},\ldots \left\{S_{M-1},T_{M-1},U_{M-1}\right\},\;\left\{S_{M},T_{M},U_{M}\right\}\right\}$.

The paired sequence of two sentences in a binary classification can be represented as $P=\left\{\left\{S_{1},S_{2}\right\},\;\left\{S_{2},S_{3}\right\},\ldots \left\{S_{M-2},S_{M-1}\right\},\;\left\{S_{M-1},S_{M}\right\}\right\}$. The addition of a vector T of speech intention tags to the series of dialogue sentence pairs can be denoted as $P^{\prime }=\left\{\left\{S_{1},S_{2},T_{1,2}\right\},\;\left\{S_{2},S_{3},T_{2,3}\right\},\ldots \left\{S_{M-2},S_{M-1},T_{M-2,M-1}\right\},\;\left\{S_{M-1},S_{M},T_{M-1,M}\right\}\right\}$.

If we add a vector U representing the speaker type (interviewee or interviewer), we obtain $P^{\prime \prime }=\left\{\left\{S_{1},S_{2},T_{1,2},U_{1,2}\right\},\;\left\{S_{2},S_{3},T_{2,3},U_{2,3}\right\},\ldots \left\{S_{M-2},S_{M-1},T_{M-2,M-1},U_{M-2,M-1}\right\},\;\left\{S_{M-1},S_{M},T_{M-1,M},U_{M-1,M}\right\}\right\}$.

The prediction target *Y* can be expressed as $Y=\left\{y_{1,2},y_{2,3},\ldots ,y_{M,M-1}\right\}$. In a multi-task learning model, the prediction target (output) *Y* is represented in two ways: $Y^{\prime }=\left\{\left\{y_{1,2},t_{1},t_{2}\right\},\left\{y_{2,3},t_{2},t_{3}\right\},\ldots ,\left\{y_{M-1,M},t_{M-1},t_{M}\right\}\right\}$ (topic break label; *t*, *n*, and speech intention tag) and $Y^{\prime \prime }=\left\{\left\{y_{1,2},t_{1},t_{2},u_{1},u_{2}\right\},\left\{y_{2,3},t_{2,},t_{3},u_{2},u_{3}\right\},\ldots ,\left\{y_{M-1,M},t_{M-1},t_{M},u_{M-1},u_{M}\right\}\right\}$ (topic break label; *nt*, *tn*, *nn*, speech intention tag, and speaker type).

Figure 4 shows the architecture of the neural networks for multi-input, triple-task learning. Let *t* be the label of a topic break utterance, and *n* be the label for a non-topic-break utterance. In the case of three-valued classification, there are three possible labels, *nt*, *tn*, and *nn*, depending on the order of t and n. Situation tt, wherein there are consecutive topic breaks, is not included in this study because it does not exist in the actual corpus and is inappropriate as a break. In the architecture shown in this figure, layers 1, 2, and 3 have dropout rates of 0.4, 0.1, and 0.1, respectively.

In multi-task learning, a shared layer facilitates the sharing of weights of relevant features between each prediction task. Although multi-task learning is sometimes used in machine learning algorithms other than neural networks, the advantage of “sharing weights” is more apparent in neural networks. In our method, the sentence vector, intention vector, and speaker vector used as inputs are all interrelated elements. Therefore, we place a shared layer just before the output layer in the proposed model and trained a classification model efficiently and accurately by concatenating the features. Although it is possible to construct a network with concatenated features, the number of dimensions of each feature is too large; thus, the model is designed with multiple inputs.

The Keras module of Tensorflow (Tensorflow. keras (https://www.tensorflow.org/api_docs/python/tf/keras, accessed on 16 January 2022)) was used to build and train the neural network.

## 5. Experiments

In order to evaluate the effectiveness of the proposed method, we conducted experiments under several conditions (with tags, without tags, multi-tasking (topic breaks and speech intention tags), triple-tasking (topic breaks, speech intention tags, and speaker labels), and single-tasking). As a baseline method, we compared our approach with SVM, logistic regression, and random forest. In addition to BERT(CLS), DistilBERT, and Sentence BERT, we also used the TF-IDF weighted bag-of-words vector (tf-idf vector) as features for comparison.

### 5.1. Experimental Condition for Evaluation Experiment

The parameters for training the machine learning models, such as neural networks and SVMs, are summarized in Table 5.

SVM, logistic regression, and random forest were performed using scikit-learn’s class library, and TF-IDF was calculated using scikit-learn’s TFIDFvectorizer (https://scikit-learn.org/stable/modules/generated/sklearn.feature_extraction.text.TfidfVectorizer.html, accessed on 16 January 2022). Training and testing were performed using a five-part cross-validation method. In the training step, the training data and validation data were split at a ratio of 8:2. Two types of evaluations were used: accuracy and F1-score of topic-separated labels. Given that there is a large bias in the number of data depending on the class, we used random undersampling to adjust the number of labels in the training and test data such that they were almost the same.

The expression for determining accuracy is shown in Equation (4). The set of classes *C* can be represented as $C=\left\{c|t,n\right\}$ for binary classification and $C=\left\{c|nt,tn,nn\right\}$ for ternary classification. 

*M_cc_* indicates the number of cases of class *c* that is correctly predicted as class c by the prediction model. *n* is the number of cases, and *M_ij_* is the sum of the number of cases of class *i* that is predicted as class *j* by the predictive model, that is, the total number of cases. 

Equations (5) and (6) show the formulas for recall and precision, respectively, for positive examples (*t* labels for binary classification, *nt*, and *tn* labels for three-valued classification). Equation (7) shows the formula for the F1-score. *TP_c_* is the number of cases of class *c* that is correctly predicted by the prediction model. *FP_c_* is the number of non-class *c* cases that are incorrectly predicted as class *c* by the prediction model. *FN_c_* represents the number of non-class *c* cases that are correctly predicted as classes other than *c* by the prediction model.
(4)$$Accuracy=\frac{\sum_{c=1}^{n}M_{cc}}{\sum_{i=1}^{n}\sum_{j=1}^{n}M_{ij}}$$
(5)$$\begin{array}{cc} {\mathrm{Recall}}_{c} & =\frac{TP_{c}}{TP_{c}+FN_{c}}\\ & =\frac{M_{cc}}{\sum_{j=1}^{m}M_{cj}} \end{array}$$
(6)$$\begin{array}{cc} {\mathrm{Precision}}_{c} & =\frac{TP_{c}}{TP_{c}+FP_{c}}\\ & =\frac{M_{cc}}{\sum_{j=1}^{n}M_{jc}} \end{array}$$
(7)$$\begin{array}{cc} {F1-\mathrm{score}}_{c} & =\frac{2}{\frac{1}{{\mathrm{Recall}}_{c}}+\frac{1}{{\mathrm{Precision}}_{c}}}\\ & =\frac{2TP_{c}}{2TP_{c}+FN_{c}+FP_{c}} \end{array}$$

### 5.2. Experimental Results

Table 6 shows the accuracy of the neural network, and Table 7 shows the highest value of F1-score (topic breaks) for each task (singletask, tripletask, and multitask). Table 8 and Table 9 show the results of the SVM, logistic regression, and random forest, respectively.

The results show that adding a speech intention tag to the features improves the accuracy. It was also determined that the neural network architecture exhibited superior performance in the case of topic-separated binary classification compared to algorithms such as SVM, logistic regression, and random forest. However, under some conditions, using the tf-idf vector as a feature yielded better results than other features, such as BERT.

Output “bin” and “tri” in the table indicates the types of classification tags to be evaluated. They represent binary topic break tags (t, n) for bin and ternary break tags (nt, tn, nn) for “tri.” “t” is a topic break utterance, and “n” represents other utterances. In the case of task, “triple” represents the learning of three types of predictive tags. Specifically, it represents the topic break tag, speech intention tag, and speaker type (interviewer or interviewer). “multi” represents multi-task learning with topic break tags and speech intention tags as prediction targets. “single” indicates single-task learning, wherein only topic breaks are used for prediction.

## 6. Discussions

In the proposed method, when the speech intention tag is not used as a feature, the performance of the multitask model is higher compared to that of the existing single-task model. The performance of the proposed method was higher for binary classification. In terms of the F-score, the multi-task model with binary classification showed the highest accuracy of 0.84 when DistilBERT was used as a feature and the speech intention tag was added.

For three-valued classification, without the speech intention tag, all methods achieved an accuracy of nearly 70%, and with the speech intention tag, an accuracy of 0.75 was obtained.

However, SVM, random forest, and logistic regression, which were used as baseline methods, also achieved the same maximum accuracy (0.81) using speech intention tags. It can be concluded that the advantages of neural networks and advanced sentence embedding, such as BERT, were not fully exploited.

In an interview dialogue system, the intention of the interviewee is known; thus, there are no issues associated with the use of the intention tag as a feature. However, when the intention tag is not known in advance, such as when the interviewer does not participate in the conversation, a multi-task model can be useful because it can detect topic segmentation with higher accuracy than a single-task model. When topic segmentation is performed on dialogue history data from different dialogue systems, it is highly likely that information on speech intentions will not be available; thus, models based on multi-task learning can be used.

In this study, undersampling was used to reduce the bias of the training data in evaluation experiments, which may have resulted in insufficient training. Random oversampling and synthetic minority oversampling technique (SMOTE) [59] are two methods of oversampling to match the number of data between classes. SMOTE is an algorithm that creates new feature vectors. Therefore, in the case of multiple inputs, it is necessary to concatenate the features and divide them into multiple inputs before training.

In addition, in the case of multi-task learning, the combination pattern of output vectors becomes large, and it becomes difficult to adjust the imbalance between classes. There are several methods of data expansion in natural language processing, such as replacing words in sentences with similar words. One problem with this method is that sentence meaning is not necessarily established after the substitution of similar words. Word substitution is also possible using BERT’s mask language model, but it is thought to be plagued by the same problem as a similar word substitution.

Here, we analyzed which words are effective features for topic segmentation when tf-idf is used as a feature by using recursive feature elimination (RFE) for feature selection. When the maximum of 100 features was selected by RFE, Table 10 shows the results of comparing the features selected in each of the five trials using three different algorithms: SVM, random forest, and logistic regression. We used tf-idf as a feature because using an embedding such as BERT obscures the importance of each word and does not allow for good feature selection. Finetuning of BERT could be used to determine the important words and speech intention tags that contribute to the classification by calculating the weight of attention. However, when neural network training is used, it is affected by the dropout of neurons and shuffling in batch selection during trials. In this respect, we adopted tf-idf because it can take into account the importance of each word, and we thought that the combination of SVM (or logistic regression and random forest) and RFE would make it easier to interpret the results of feature selection.

Bold letters in the table are commonly selected features among the models. From the results, we can observe that, as word features, many expressions are frequently used by interviewers. However, features such as <s> and <par> are effective speech intention tags. Using the features of the utterance labels, we obtained a maximum accuracy of 83% when predicting using only the features selected by RFE with random forest. This accuracy is comparable to that of a multi-task neural network with DistilBERT and speech intention tags. Given that the accuracy was improved by reducing the number of features to 100 dimensions by feature selection, it can be assumed that the unknown words had little effect on accuracy reduction.

In many cases, the topic break is caused by the interviewer who takes the initiative in the dialogue. This indicates that topic breaks can be detected 80% of the time, because we were able to identify the interviewer and examine intrinsic features (such as speech intention tag).

## 7. Conclusions

In this report, we proposed a topic segmentation method for interview dialogue to realize an interview dialogue system that can efficiently and flexibly elicit information from a dialogue partner. However, the topic of the dialogue changes gradually; thus, it is difficult to predict from back and forth relations. The proposed method focuses on whether the speaker is an interviewer or not and uses the speech intention tags of the interviewer’s utterance as a feature. This method can be applied to two consecutive utterances. The accuracy of this method is higher than that of three-valued classifying pairs of two consecutive utterances as features. Thus, it was possible to clarify the effectiveness of speech intention tags in an interview dialogue.

By using this method, it is easy to detect topic breaks in dialogue when the interviewer takes the initiative. The application to dialogue systems can be used to understand the topic transition status and topic content.

In future work, we plan to create a model for estimating topic vectors from topic-delimited dialogue segments by improving existing methods and to construct a model for generating speech from similar topics. In addition, we intend to further expand the corpus and use the trained model to detect similar topic segments to realize an interview dialogue system that can discern the user’s interest and enhance the user’s charm and characteristics.

## Figures and Tables

**Figure 1 sensors-22-00694-f001:**
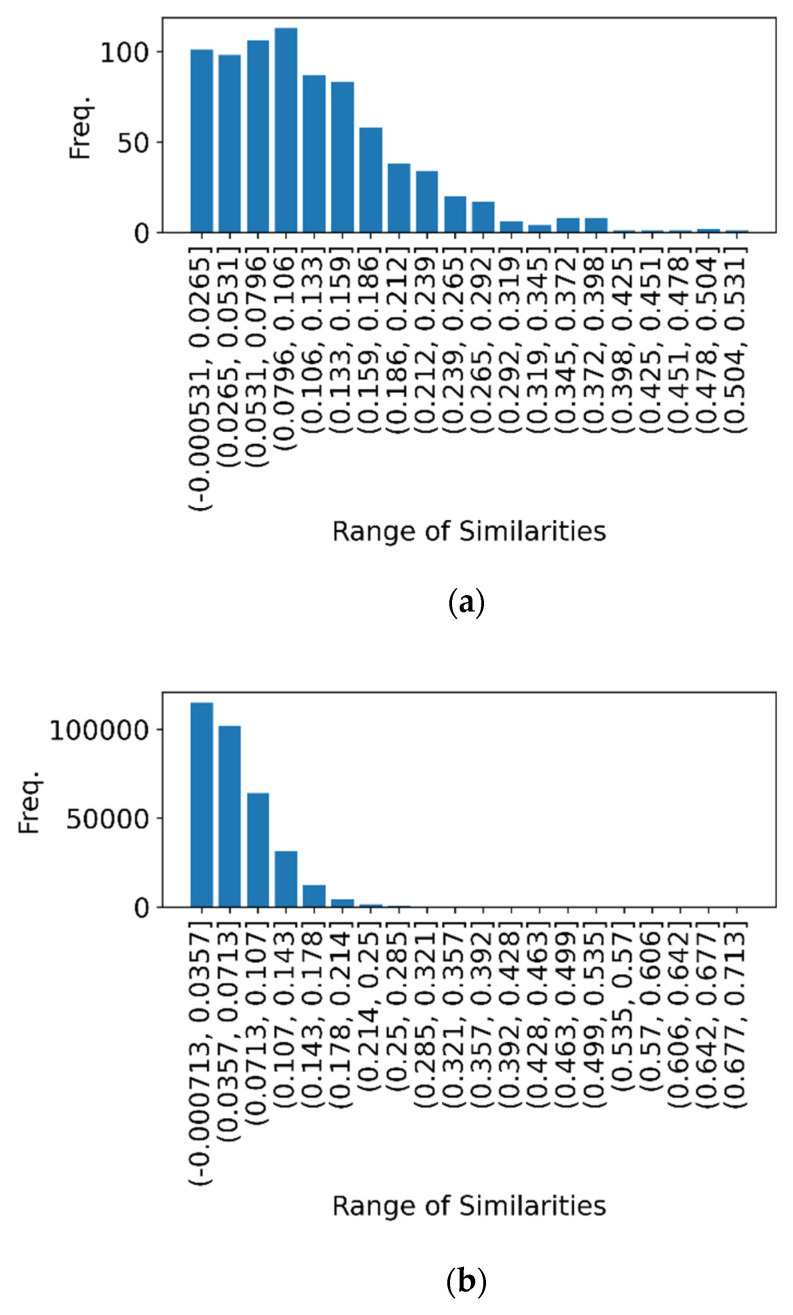
Distribution of similarity for topic sections. (**a**) Histogram of similarity for adjacent topic sections, (**b**) Histogram of similarity for all topic sections.

**Figure 2 sensors-22-00694-f002:**
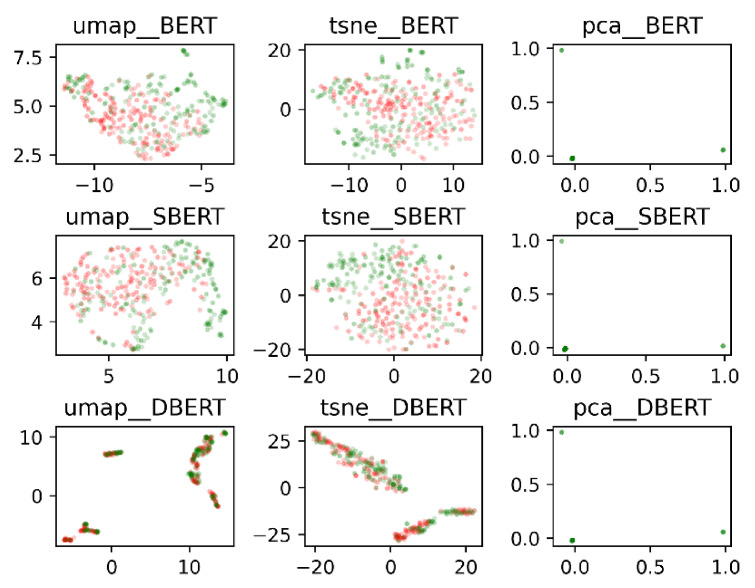
Two-dimensional plot of feature vector; BERT, SBERT, and DBERT.

**Figure 3 sensors-22-00694-f003:**
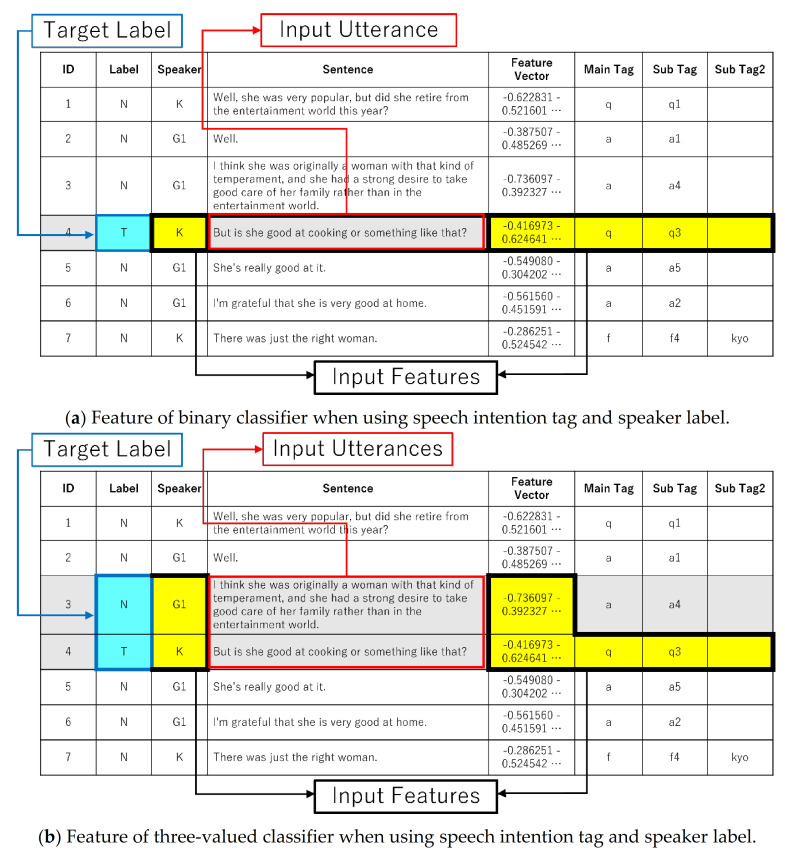
Input features for binary/three-valued classifier.

**Figure 4 sensors-22-00694-f004:**
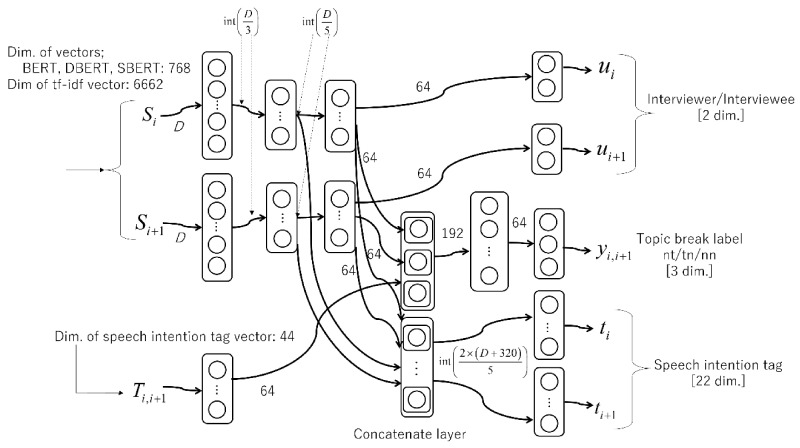
Triple-task multi-input neural networks for dialogue topic segmentation task.

**Table 1 sensors-22-00694-t001:** Speech intention tag.

Main Tag	Sub-Tag	Sub-Tag 2
“q”: Question	Pre-arranged question (“q1”)	Nest (“nest”)
	Opinion (“q2”)	
	Others (“q3”)	
“a”: Answer	Agreement/Disagreement (“a1”)	Nest (“nest”) Self-remark (“j”)
	Self-disclosure (opinion) (“a2”)	
	Self-disclosure (others) (“a3”)	
	Explanation of another person (opinion) (“a4”)	
	Explanation of another person (others) (“a5”)	
“f”: Feedback	Compliment (“f1”)	Empathy (“kyo”)
	Help (“f2”)	
	Asking back/Repetition (“f3”)	
	Others (“f4”)	
“c”: Comment	Compliment (“c1”)	Empathy (“kyo”)
	Others (“c2”)	
“b1”: Explanation to audience		
“b2”: Filler		
“b3”: Laughter		
“b4”: Greeting		

**Table 2 sensors-22-00694-t002:** Breakdown of interview dialogue corpus.

Main Tag	Interviewer	Interviewee
q	2319	151
a	337	6442
f	2334	393
c	1548	3
b1	401	6
b2	42	96
b3	173	118
b4	160	238

**Table 3 sensors-22-00694-t003:** Breakdown of topic breaks for each utterance speaker.

Speaker	Label (t)	Rate	Label (n)	Rate	Label(t)+Label(n)
Interviewer	753	0.10	6749	0.90	7502
Interviewee	84	0.01	7684	0.99	7768

**Table 4 sensors-22-00694-t004:** Comparison of word sets between similar adjacent topics.

Similarity	t-1	t
0.35	player, **violin**, pressure, offer, strong, honor, dad, feeling	**violin**, musical performance, very, exception, wind instrument, drums, keyboard
0.34	son, **mother**, of course, think, good, that, not	language, but, news, anxiety stage, group, **mother**

**Table 5 sensors-22-00694-t005:** Training parameters for each model.

Model	Parameters	Value
Neural Networks	epochs	5–100
	optimizer	Adam
	loss function	categorical crossentropy
SVM	kernel	rbf
	regularization parameter C	1.0
	class weight	balanced
Logistic Regression	default parameters	
Random Forest	default parameters	

**Table 6 sensors-22-00694-t006:** Experimental results (accuracy).

Tag	Output (bin/tri)	Task	Feature	Epochs	Accuracy
without	bin	triple	DBERT	70	**0.79**
		multi	DBERT	90	**0.79**
		single	DBERT	30	0.78
	tri	triple	DBERT	90	**0.70**
		multi	BERT	20	0.69
		single	DBERT	40	0.69
with	bin	triple	DBERT	20	0.83
		multi	DBERT	20	**0.84**
		single	DBERT	10	0.83
	tri	triple	DBERT	20	**0.76**
		multi	DBERT	20	**0.76**
		single	BERT	10	0.75

**Table 7 sensors-22-00694-t007:** Experimental results (F1 of true label).

Tag	Output (bin/tri)	Task	Feature	Epochs	F1-score
without	bin	triple	DBERT	70	**0.78**
		multi	DBERT	80	**0.78**
		single	DBERT	50	0.77
	tri	triple	BERT	30	**0.74**
		multi	BERT	20	0.73
		single	BERT	5	0.73
with	bin	triple	DBERT	20	**0.83**
		multi	DBERT	20	**0.83**
		single	DBERT	20	**0.83**
	tri	triple	DBERT	20	0.785
		multi	DBERT	30	**0.795**
		single	DBERT	20	0.79

**Table 8 sensors-22-00694-t008:** Accuracy of baseline methods.

Tag	Output (bin/tri)	Model	Feature	Accuracy
without	bin	SVM	BERT	**0.78**
		Logistic Regression	BERT	0.76
		Random Forest	BERT	0.78
	tri	SVM	BERT	**0.75**
		Logistic Regression	BERT	0.75
		Random Forest	BERT	0.74
with	bin	SVM	BERT	**0.83**
		Logistic Regression	TFIDF	0.82
		Random Forest	TFIDF	0.81
	tri	SVM	BERT	**0.82**
		Logistic Regression	BERT	**0.82**
		Random Forest	BERT	0.81

**Table 9 sensors-22-00694-t009:** F1-score of baseline methods.

Tag	Output (bin/tri)	Model	Feature	F1-score
without	bin	SVM	BERT	**0.79**
		Logistic Regression	BERT	0.76
		Random Forest	BERT	0.79
	tri	SVM	BERT	**0.76**
		Logistic Regression	BERT	0.75
		Random Forest	BERT	0.72
with	bin	SVM	BERT	**0.83**
		Logistic Regression	DBERT	0.82
		Random Forest	BERT	0.80
	tri	SVM	BERT	**0.82**
		Logistic Regression	BERT	**0.82**
		Random Forest	BERT	0.80

**Table 10 sensors-22-00694-t010:** Comparison of selected features.

Feature	SVM	Random Forest	Logistic Regression
tf-idf	***dou*, *rassharu*, you, *irassharu*, today**, graduation, change, **guest**, mother, now, ***gurai*, marriage, *san*, *da*, but**, etc.	***dou*, *rassharu*, *irassharu*, you, today**, yes, actor, aunt, **guest**, mother, ***gurai***, *kedo*, **marriage**, this, ***san***, Kuroyanagi, great, That’s right, that, it, ***da*, but**, etc.	***dou*, *rassharu*, *irassharu*, you, today**, yes, actor, **guest, *gurai*, marriage, say, *san***, great, That’s right, such, ***da*, but**, etc.
tf-idf + speech intention tag vector	you, *bera*, high, nice, **<s>**, **<par>**, <nest>, etc.	*dou*, yes, *rassharu*, *irassharu*, today, guest, mother, gurai, marriage, this, *san*, Kuroyanagi, That’s right, *da*, but, <a>, <d>, <k>, <q>, **<s>**, <else>, <iken>, <jik>, **<par>**, <yoi>, etc.	yes, everyone, you, *kashira*, That’s right, it, *da*, but, <k>, <q>, **<s>**, **<par>**, <yoi>, <kyo>, <nest>, etc.

## Data Availability

Not applicable.

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
