# Peer review of "Topic Break Detection in Interview Dialogues Using Sentence Embedding of Utterance and Speech Intention Based on Multitask Neural Networks"

_sensors, 2022, doi:10.3390/s22020694_

Round 1
Reviewer 1 Report
The following comments must need to be addressed in the revised version of the paper:
1. The abstract needs to be logically divided into the sections: background, existing literature, problem statement, [proposed methodology, results and practical implications.
2. The introduction needs to enumerate the limitations of the existing studies in the introduction section.
3. Contributions of the proposed study need to be highlighted at the end of introduction section.
4. Main sections of the paper must be followed by the introductory lines about the section and should not directly jump to the sub-section. For example: 2. Related Works ; 2.1. Studies on Dialogue System
5. Figure 1 needs improvement. In the current form its visibility is very low.
6. Grammar needs improvement.
Reviewer 2 Report
This article presented a method for recognizing topic breaks in dialogue to achieve flexible topic switching in an interview dialogue system. The developed method seems interesting to me.
Below are my suggestions for improving the paper's overall quality and its structures.
- Abstract: The start of the abstract should be from the background statement, then described issues in the existing system/method, then proposed method importance.
- Related works 2.1:
- In line- 96, the author mentioned "there is a lot of research on dialogue systems," referencing these dialogue systems appropriately.
- Same in line 101, the author again mentioned that "there are many studies," which studies?
- Related works 2.2:
- The author used informal language, "She detects", it is best practice to avoid third-person
- In the related work, most of the existing studies deal with multi-persons dialogues. However, your work deals with two-person dialogues seem that the existing studies try to resolve complex problems compared to your work, which needs more justification.
- In the data annotation section, manually labeled data ware verified with two other researchers,
- what is the agreement's significant value?
- Which method (i.e, Kappa statistics and Kendall's coefficients) did the author use for a mutual agreement?
- Why the author uses the TF-IDF vectorization instead of another embedding vector for corpus analysis?
- The last section heading number should be 7 instead of 5.
Round 2
Reviewer 1 Report
All my concerns are addressed.